# Pattern of seasonal variation in rates of predation between spider families is temporally stable in a food web with widespread intraguild predation

**David H. Wise**[1]*, **Robin M. Mores**[1], **Jennifer M. Pajda-De La O**[2], **Matthew A. McCary**[3]

**1** Department of Biological Sciences, University of Illinois, Chicago, Illinois, United States of America,
**2** Department of Mathematics, Statistics, and Computer Science, University of Illinois, Chicago, Illinois, United States of America, **3** Department of BioSciences, Rice University, Houston, Texas, United States of America

* mm160@rice.edu

## Abstract

Intraguild predation (IGP)–predation between generalist predators (IGPredator and IGPrey) that potentially compete for a shared prey resource–is a common interaction module in terrestrial food webs. Understanding temporal variation in webs with widespread IGP is relevant to testing food web theory. We investigated temporal constancy in the structure of such a system: the spider-focused food web of the forest floor. Multiplex PCR was used to detect prey DNA in 3,300 adult spiders collected from the floor of a deciduous forest during spring, summer, and fall over four years. Because only spiders were defined as consumers, the web was tripartite, with 11 consumer nodes (spider families) and 22 resource nodes: 11 non-spider arthropod taxa (order- or family-level) and the 11 spider families. Most (99%) spider-spider predation was on spider IGPrey, and ~90% of these interactions were restricted to spider families within the same broadly defined foraging mode (cursorial or web-spinning spiders). Bootstrapped-derived confidence intervals (BCI's) for two indices of web structure, restricted connectance and interaction evenness, overlapped broadly across years and seasons. A third index, % IGPrey (% IGPrey among all prey of spiders), was similar across years (~50%) but varied seasonally, with a summer rate (65%) ~1.8x higher than spring and fall. This seasonal pattern was consistent across years. Our results suggest that extensive spider predation on spider IGPrey that exhibits consistent seasonal variation in frequency, and that occurs primarily within two broadly defined spider-spider interaction pathways, must be incorporated into models of the dynamics of forest-floor food webs.

## Introduction

Mathematical models of food web structure and dynamics ultimately must be tested by comparing model predictions with the behavior of natural food webs. Because models frequently assume an equilibrium state, one would like to find natural food webs at equilibrium for

**Data Availability Statement:** Raw data used to construct the interaction matrices and the R code for bootstrapping margins of error for restricted connectance and interaction evenness are given in

GitHub (https://github.com/mmccar26/spider-focused-food-web) and have been archived using Zenodo with DOI xxx.xxxx.

**Funding:** The author(s) received no specific funding for this work.

**Competing interests:** The authors have declared that no competing interests exist.

making comparisons. Determining whether an ecological system is at equilibrium is a contentious issue. It is reasonable to argue, though, that the natural food web being compared should exhibit some degree of temporal constancy in structure; or if variable, a pattern of variability that is predictable, such as seasonal changes in structure. Unfortunately, empirical food webs frequently are summary webs constructed from pooled seasonal or yearly data, which has prompted calls to obtain more information on the temporal constancy and variability in the structure of natural food webs [1, 2]. Identifying empirical food webs that appear to be at equilibrium over several years is critical to testing theories about factors that affect food web stability. One factor hypothesized to influence web stability is the prevalence of trophic-level ominivory.

The impact of trophic-level omnivory on the stability of food webs has been a focus of theoretical and empirical investigations for many years [3]. The simplest example of trophic-level omnivory is intraguild predation (IGP) [4], defined most broadly as a module [5] in which two generalist predators that share a potentially limiting resource on a lower trophic level (a herbivore, microbivore, detritivore or another predator) also feed upon each other. IGP can be "asymmetrical" (a module in which a "top" generalist predator, the IGPredator, preys upon an IGPrey, but not vice-versa); or "reciprocal", in which the two generalist predators prey upon each other, usually at different intensities. IGP is widespread in nature [6], but quantitative estimates of IGP in most systems are few [7], particularly for predaceous arthropods [8]. Knowledge of changes over time in the pattern of IGP in a food web is even more sparse [9]. Thus, understanding the degree of temporal constancy in the quantitative structure of food webs with high degrees of IGP will contribute to linking food web theory to the structure and dynamics of natural food webs.

We investigated the constancy of three structural indices in a food web that exhibited extensive IGP. Our first two indices, "connectance" and "interaction evenness", measure somewhat abstract features of the overall network. Connectance is the proportion of potential linkages that are realized, and the version of interaction evenness that we employed summarized how feeding intensities from generalist consumer nodes are spread across resource nodes. Changes in our third index of web structure, "% IGPrey" (% of all prey in the focal consumer that are IGPrey), could reflect changes in relative energy flow through the two channels of IGP modules. The research reported here examines the temporal stability of these three indices in a widely distributed terrestrial food web.

Our research studied a spider-focused food web on the floor of a deciduous forest. Most aboveground net primary production in forests falls to the forest floor where it becomes a major source of input to the detritus-based food web of the litter horizon of the soil subsystem [10]. In the litter layer, interactions between arthropod predators and their prey can generate trophic cascades that indirectly affect litter decomposition and nutrient cycling; and changes in the resource base can exert bottom-up control of predator densities [11, 12]. Spiders are ubiquitous, species-diverse and abundant on the forest floor [13, 14]. They are generalist predators that feed upon a wide spectrum of prey, including other spiders [15–17]. Thus, we expected widespread IGP among spiders to be a major feature of the forest-floor web we selected to study.

Because only spiders were defined as consumers in our spider-focused food web, it is a tripartite web with 11 consumer nodes (spider families) and 22 resource (prey) nodes: the 11 spider families and 11 non-spider arthropod taxa defined at a broad scale (order or family). Similarity in foraging behavior within a spider family makes this level of resolution reasonable [18], and the broad prey spectrum of most spider families justifies a broad definition of the non-spider prey nodes. This web is a hybrid of the classical "sink" and "source" food webs [19] due to the potential importance of reciprocal IGP. Defining spiders as the top consumers and

focusing on their prey makes the web a "sink" web. Because all spiders can potentially be the prey of all other spiders (i.e., reciprocal IGP), the web also is a partial "source web".

Although predation between two generalist predators sometimes is described loosely as "intraguild predation," we use the strict definition of IGP and do not describe all spider-spider predation as examples of IGP. In our tripartite web, predation between two spider families was IGP only if the two families shared a non-spider prey taxon. Thus, to exclude spider-spider predation events that were not part of an IGP module, we defined the prevalence of IGP as the % of all prey that were IGPrey in spider diets (i.e., % IGPrey).

We had no strong *a priori* hypotheses regarding the temporal variability of connectance, interaction evenness, or % IGPrey, but we did suspect that the first two would be the least variable because the nodes were broadly defined. For example, connectance in webs with "trophic-species" nodes is less sensitive to sampling effort than connectance in webs where nodes are species [20]. However, Cardoso *et al*. [18] argue that spider families "may potentially serve as ecological surrogates for species," which suggests that temporal variation in patterns of trophic connections in our study system could result in detectable changes in connectance. Arguing against this expectation is the absence of specialist spiders in our web and the taxonomic diversity of the non-spider prey nodes–properties that made it reasonable to expect that both connectance and interaction evenness would exhibit considerable constancy.

Making a solid prediction about the temporal constancy of % IGPrey in spider diets is more challenging. For example, the abundance of alternative non-spider prey can impact rates of predation on IGPrey [21]. Abundances of alternative prey might change predictably over the growing season, which could lead to a predictable change in the frequency of feeding by spiders on other spiders. However, the impact of swings in abiotic factors, such as rainfall, could alter the activity and/or resource base of alternative prey, leading to unpredictable changes in the seasonal pattern of feeding on IGPrey. Such variability, coupled with the impact of population size structure on the dynamics of IGP [4], and the numerous IGP modules subsumed within modules defined at a broad taxonomic level, weakened any concrete predictions that we might have made about temporal constancy in the pattern of IGP.

We uncovered trophic interactions using multiplex PCR [22] to screen spiders simultaneously for DNA from a wide range of potential prey. To assess temporal variation in food web structure, we analyzed enough adult spiders (3,300) to evaluate changes in web structure for three seasons (spring, summer and fall) over four years. We first construct a summary web for the entire data set and then examine sub-webs (summary webs with seasons or years pooled) to assess temporal variability across years and seasons, respectively.

## Materials and methods

### Study system

We collected spiders and potential non-spider prey from the oak-dominated (*Quercus alba* and *Q. rubra*) Swallow Cliff Woods (41˚ 40.519' N, 87˚ 51.437' W) within the 320-ha Swallow Cliff nature preserve in Palos Township, Illinois (USA). The preserve, which is within the Chicago metropolitan region, is managed by the Cook County Forest Preserve District. Forests in this region are actively managed for several invasive plants [23], and the forest floor at Swallow Cliffs contains a thick leaf-litter layer with an abundant and diverse arthropod community.

### Collecting spiders and non-spider prey

Our goal was to search the ground layer and low understory as thoroughly as possible, so that we would collect enough spiders from less-abundant families to yield the same number of spiders per family analyzed for prey DNA. We did not estimate spider densities. All collections

were made between 1000 and 1600 hours. We collected from a different location each day. The size of the area searched each day was not measured and varied with the number of searchers. Collecting areas were widely distributed throughout Swallow Cliff Woods, but we did not subdivide the Woods into sampling regions. Most terrain was upland forest, but some collections were taken from a few scattered wet/marshy areas. The number of collecting days in each season was spring (31), summer (33), and fall (29) over the years 2009, 2010, 2011 and 2012; the number of days per year was 33, 12, 34 and 14, respectively.

On each collecting day, we used both litter sifting and simple searching to capture spiders from several microhabitats. For litter sifting, we placed litter collected by hand into a flat tray (58 cm x 17 cm x 15 cm) with a screen bottom. This tray was shaken over a second tray of the same size with a solid bottom, allowing arthropods to fall through the screen to be collected by hand or aspirator. Sifted litter was returned to its original location. Spiders were also collected by hand from the litter surface, open areas in the litter, logs, low vegetation up to ~1m, and tree trunks up to ~2m. Individual spiders were placed in separate labelled vials.

Of the spiders that were eventually analyzed for prey DNA (see below), 81% were captured from either leaf litter (70%) or adjacent bare ground/logs (11%). Thus, most spiders were collected from the litter layer broadly defined. The litter layer is a fairly distinct subsystem with respect to rates of migration of arthropod predators and prey [24]. Nevertheless, we did not limit our definition of the "forest floor" to the litter layer because many spiders spin webs in vegetation close to the ground. Also, some cursorial species move back and forth between the ground and lower understory vegetation and tree trunks (for example, 84% of the Corinnidae, a guild of "foliage runners" [25], were collected from leaf litter). Therefore, we also analyzed spiders that had been collected from low vegetation (10%) and tree trunks (9%).

All specimens were placed on ice within one hour of capture. On the same day, spiders collected for detection of consumed prey using PCR were taken to the laboratory where they were weighed and stored at -20˚C in a 1.5-mL microcentrifuge tube containing 95% ethanol (EtOH). Spiders and non-spider prey (see below) intended for primer development or assay optimization (see below for details) were kept alive, weighed, placed individually into 60-mL glass vials, and provided with water *ad libitum* at room temperature. Spiders were identified to family and genus using identification guides [26–29]. Voucher specimens (one adult male and female) for each genus (when available) were archived at The Field Museum (Chicago, Illinois).

Over the four years, ~14,000 spiders (juveniles and adults) from 20 families were collected. Presence of prey DNA was tested for adult spiders from 11 abundant families (those with at least 300 adults) that live primarily on the forest floor. Spiders from six of these families (Corinnidae, Gnaphosidae, Lycosidae, Pisauridae, Salticidae, and Thomisidae) do not spin webs to capture prey ("cursorial" spiders). The other five families (Agelenidae, Dictynidae, Hahniidae, Linyphiidae, and Theridiidae) are "web spinners." This dichotomy reflects basic differences in foraging behavior [16, 17], but the distinction is not absolute. The web spinners in our food web include genera of spiders that also forage for prey off their web [18].

Non-spider arthropod prey were also collected for primer development. They were not sampled quantitatively, but were simply selected due to their apparent abundance in leaf litter and/or activity just above the litter layer, and their likely occurrence in the diets of at least one spider family [15–17, 30]. Non-spider nodes of the food web were broadly defined taxonomically (at the Order level except for Gryllidae): flies (Diptera), moths/butterflies (Lepidoptera), springtails (Collembola), ants/bees/wasps (Hymenoptera), jumping bristletails (Archaeognatha), crickets (Gryllidae), pseudoscorpions (Pseudoscorpiones), harvestmen (Opiliones), beetles (Coleoptera), earwigs (Dermaptera), and pillbugs (Isopoda).

## Molecular techniques

**Primer development and optimization.** We utilized multiplex PCR to sequence DNA from at least ten spiders from each family and at least ten specimens from each non-spider prey taxon. Each spider was first starved for at least ten days to eliminate any gut-content DNA that may have been present. Specimens were then homogenized in 180 μL of phosphate-buffered saline (PBS) (Hoefer, San Francisco, CA). DNA was then extracted with a Qiagen DNEasy Tissue Kit (Valencia, CA) using the manufacturer's protocol. Upon completion of DNA extraction, the 200μL of eluate was well-mixed, separated into 20μL aliquots, and stored at -20°C until analysis.

The general arthropod primers LCO-1490 and HCO-2198 [31] were used to amplify DNA from the mitochondrial genome's cytochrome oxidase I (COI) region. Eluate from DNA extractions was amplified and sequenced by The Field Museum (Chicago, IL) or Research Resources Center (RRC) at the University of Illinois, Chicago. Sequences were used to conduct BLASTN searches following the protocol developed by [32] using the databases GenBank and BOLD (the Barcode of Life Database). Following [33], database sequences were used only if they showed ≥97% match to submitted sequences. Sequences were aligned using the CLUS-TALW or AMPLICON programs. Primers were designed with the assistance of the IDT (Integrated DNA Technologies, Coralville, IA) program PrimerQuest and tested for melting temperature and CG content using Sci-Tools OligoAnalyzer (IDT).

**Spider gut-content testing.** After a PCR assay was developed and optimized for a particular prey taxon (spider family or non-spider arthropod), frozen field-caught adult spiders were tested for the presence of the target-prey DNA. Spiders were thawed to room temperature and underwent DNA extraction and PCR amplification as described above. The entire spider was homogenized, except for the largest individuals, for which legs were removed to increase the prey/predator DNA ratio; coxae were left attached to the body when possible because spider guts often extend into the coxae [17]. The homogenate was then mixed and 4uL were added to a well (on a 96-well plate) that contained 21 uL of Master Mix. Every run also included positive, negative, and blank controls to ensure that target DNA was amplified and that no contamination existed on the run. Positive controls consisted of DNA specific to the target taxon in question, negative controls contained the PCR Master Mix without DNA template, and blank controls were created from MBG water. A sample was considered positive for target-prey DNA within the spider's gut if the Ct value of the amplification curve was above the background threshold, if the shape of the curve was sigmoidal, and if the positive and negative controls were acceptable. Samples that did not show amplification were re-analyzed using arthropod-general primers [31] before identifying them as negative results; questionable samples (low amplification or a non-sigmoidal shape) were re-tested.

For constructing the food web, adult spiders (as close as possible to 1:1:: male:female) were selected at random from genera within the most abundant families to yield 25, 50, 75, 100, or 150 per genus (roughly in proportion to relative abundances) for a total of 300 / family. Each spider was analyzed for DNA from the entire range of potential prey, i.e., the 11 non-spider arthropod taxa and the other 10 spider families. Only 17 out of 3300 spiders tested positive for two different prey items; these were treated as two separate interactions (as if there were two spiders). Only one spider tested positive for more than two prey: a thomisid tested positive for 7 different prey taxa, which was considered an outlier and was removed from the analysis (refer to S1 File for a discussion of possible causes of the low number of spiders testing positive for more than one prey item). Because of these minor adjustments, the number of spiders tested for prey in our statistical analysis ranged from 299–304 per family.

## Analysis of food web structure

**Interaction matrix.**   PCR results were translated into an 11 x 22 interaction matrix in which each row (*i*) represented a consumer, i.e. one of the 11 focal spider families; and the *i*th cell in each column (*j*) contained the number of the *i*th consumer that tested positive for DNA for one of the other spider families or one of the 11 non-spider taxa. Intra-family predation, which includes cannibalism within species, no doubt occurred in our study system, but could not be detected by PCR. Thus, matrix cells (*i,i*) by necessity = 0, and the food web that was ana-lyzed contained not 11 x 22, but 11 x (11+10) = 231 potentially detectable predator-prey interactions.

The entire matrix, with years and seasons pooled, constituted the summary web. To exam-ine temporal variability, we created interaction matrices for each year (2009, 2010, 2011, and 2012; seasons pooled) and each season (spring, summer, and fall; years pooled). We did not analyze matrices for each year x season combination because small sample sizes for many com-binations would have made such analyses to be of questionable value.

**Food web structure: Graphical depictions.**   Interactions in our tripartite food web can most easily be depicted as an interaction network using a graphical technique that places the consumer categories (the 11 spider families) in the middle row, with triangles connecting each spider family with the non-spider prey depicted in the lower row, and spider prey in the upper row [34]. These diagrams reflect the direction and magnitude of trophic interactions based solely upon spiders that tested positive for prey DNA. The width of each triangle reflects the relative proportion of that prey in the consumer's diet. Proportions of tested spiders that had no detectable prey DNA are reported separately (S1 Table).

**Food web structure: Indices.**   Numerous indices have been proposed to measure food web structure. The goal often has been to determine whether the observed index indicates a degree of organization and distribution of links between nodes that differ from a null hypothe-sis of a random arrangement. We addressed a different question: do measures of food web structure vary temporally? Three indices were calculated that are appropriate to evaluating changes in a spider-focused, hybrid source-sink web. The first two, restricted connectance and interaction evenness, are abstract measures of overall network structure. The third measures the frequency of predator-predator interactions within the context of widespread IGP.

*Restricted connectance.* Measures of connectance compare the number of realized links to the number of possible links in a food web [1, 35]. The number of possible links varies accord-ing to assumptions about the nature and direction of predator-prey interactions in the web, yielding a range of specific terms for connectance [36]. We used a definition, more restrictive than most, that (*i*) matches the hybrid sink-source structure of our spider-focused web, and (*ii*) reflects the limitations of our PCR analyses, i.e. we could not test for intra-family predation (analogous to cannibalism in a food web in which species are nodes). To make this distinction clear, we use the term *restricted connectance*, defined as:

$$C_R = \frac{L}{J(J-1) + JI}$$

where L is the number of realized links, J is the number of consumers (spider families), and I is the number of non-spider prey. Restricted connectance, which can vary from 0 to 1, is calcu-lated from a binary (i.e. presence/absence) interaction matrix. We re-sampled, with replace-ment, from subsets of the full data set to estimate margins of error (95% BCI) for yearly and seasonal estimates of restricted connectance. From 5,000 simulated values we extracted the median, and the 2.5 and 97.5 percentiles for the bootstrapped confidence interval (95% BCI). Analyses were performed in R version 3.2.1. For the yearly webs, three families (Corinnidae,

Pisauridae and Gnaphosidae) were removed from the full data set due to low numbers collected in 2009; 25 spiders were then sampled, with replacement, from each family for each year. Three families (Corinnidae, Pisauridae and Agelenidae) were removed before bootstrapping seasonal data due to low numbers collected in one season; 30 spiders were then sampled, with replacement, from each remaining family for each season.

*Interaction evenness.* This metric reflects the relative intensity of interactions across a quantitative food web by incorporating the actual values in each cell of the interaction matrix. The metric was calculated using the "network-level' function in the 'bipartite' R package [37]. The metric scales between 0 and 1, with 1 = maximum evenness. We employed the "sum" version of the index, which calculates the index only for realized interactions. Margins of error (95% BCI) for yearly and seasonal values of interaction evenness were estimated using the procedure employed for restricted connectance.

*% IGPrey.* A spider preyed upon by another other spider was an IGPrey if both spiders shared at least one non-spider prey, i.e., if both spiders, along with the non-spider prey, formed an IGP module. We defined % IGPrey as the percentage of all detected prey that were classified as IGPrey. Margins of error (95% CI) for values of % IGPrey by year (seasons pooled) and by season (years pooled) were based upon the normal distribution because sample sizes and success/failure rates satisfied assumptions. Whether there was an interaction between year and season was evaluated by selecting the best model for a 4 x 3 x 2 contingency table using a loglinear model with a log link function and a Poisson response [38, 39]. The analysis was carried out using SAS version 9.4.

## Results

Of 3300 analyzed spiders, 45% tested positive for prey DNA. Percent positive per family ranged from 41% to 52% among cursorial spiders and from 37% to 60% among web-spinners. Further details appear in S1 Table.

### Food web structure: Graphical depictions

**Summary web.**   Several features characterize the broad pattern of trophic interactions in the summary web (years and seasons pooled) (Fig 1): (i) The three most-consumed non-spider taxa (Diptera, Lepidoptera, and Collembola) appeared in the diets of 6–9 spider families. (ii) The eight less-commonly consumed non-spider taxa (Hymenoptera, Archaeognatha, Gryllidae, Pseudoscorpiones, Opiliones, Coleoptera, Dermaptera, and Isopotera) appeared almost exclusively in the diets of cursorial spiders. (iii) All spiders except Agelenidae appeared in the diets of 2–6 other spider families; no spiders fed on agelenids. (iv) Overall rates of spider-spider predation were high. (v) Spider-spider predation occurred primarily within, not between, the two broad foraging categories of cursorial and web-spinning spiders. Cursorial spiders fed primarily on other cursorial spiders, and web spinners preyed largely on other families of web spinners. A more detailed analysis of the summary web appears in S2 Table.

### Yearly webs

Overall patterns of trophic interactions, including proportions of intraguild prey, appear to be similar across years (Fig 2). Large differences between years in number of consumers depicted, such as those between 2009 and 2011 for Lycosidae and Pisauridae, may largely reflect differences in searching success. Research assistants varied between years, and the more visible families were likely collected earlier in the research. For example, lycosids (wolf spiders), which were heavily collected the first year, are the most conspicuous litter spiders because they are abundant and are very active on the litter surface and open ground.

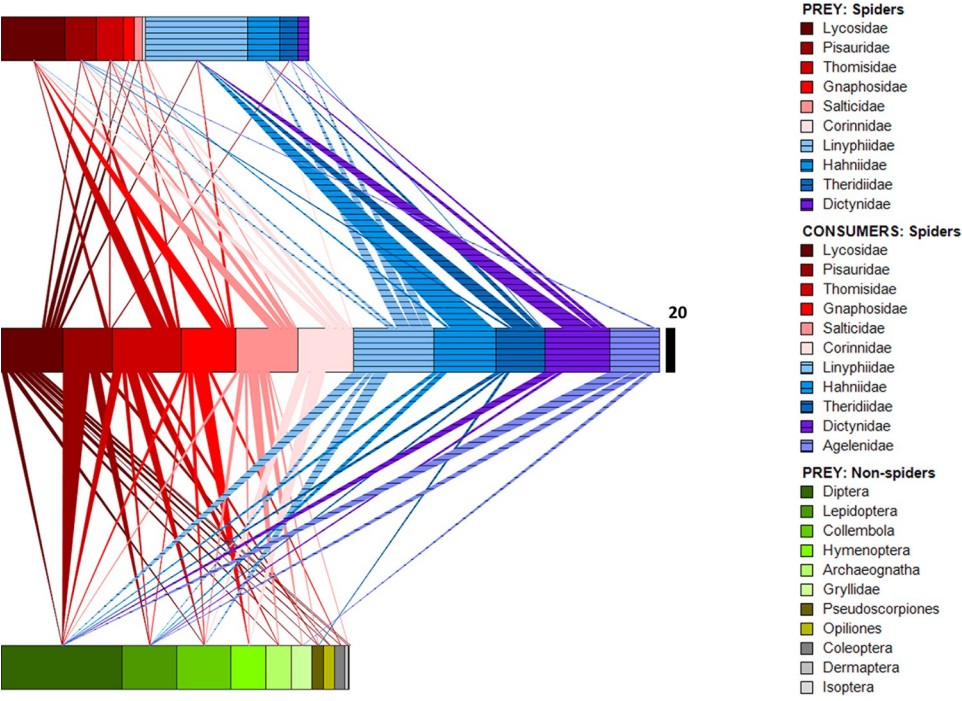

**Fig 1. Summary food web (years, seasons combined).** Middle Panel: CONSUMERS (spider families). Solid bars are cursorial spiders, bars with horizontal lines are web spinners. Relative widths of triangles connecting a CONSUMER to a PREY category reflects percentages of prey in the CONSUMER's diet. Width of a CONSUMER bar indicates number of prey items detected for that family (black scaling bar = 20; this scale also applies to top and bottom bars). Number of prey items detected per spider family ranged from 110 to 182. Total number detected = 1488. Top Panel: Spiders that were PREY, total number = 698. Bottom Panel: Non-spider PREY, total number = 790.

## Seasonal webs

Changes in the pattern of trophic connections appear to be more pronounced between seasons than between years (Fig 3). The most obvious variation was the seasonal shift in the frequency of spider-spider predation.

Numbers of analyzed spiders (all were adults) also differed between seasons for some families. The clearest seasonal difference in the number of spiders analyzed for prey DNA was shown by the Agelenidae, which displayed a distinctly annual phenology. This family was represented by a single genus (*Agenelopsis*; S1 Table). Three other families (Corinnidae, Gnaphosidae, and Pisauridae), which were represented primarily by two or three genera (S1 Table), exhibited less-pronounced seasonal changes in numbers analyzed. Seasonal changes in trophic patterns of these latter three families resembled those of the seven more evenly represented families.

## Food web structure: Indices

**Restricted connectance.** The summary web exhibited 83 links, yielding a restricted connectance of 0.36; thus, about a third of the potentially detectable links were realized over the entire study. The food web's binary structure did not differ substantially across years, judging by the broad overlap of the boot-strapped confidence intervals around median values of restricted connectance (Fig 4A). The median value of restricted connectance appeared to increase slightly from spring to fall (Fig 4B), but evidence is weak given the degree of overlap of the 95% BCI's. Median values for years and seasons ranged from 0.22 to 0.27, which are

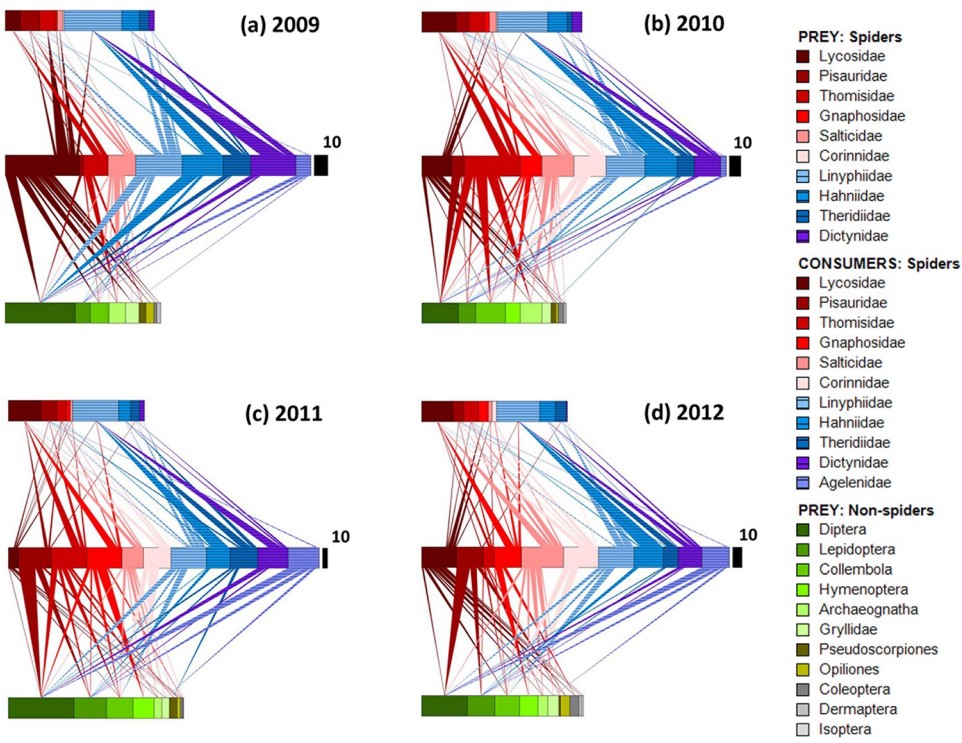

**Fig 2. Food webs for years 2009 through 2012 (seasons combined).** Key as in Fig 1, except black scaling bar represents 10 individuals. The overall length of the middle panel does not vary between years; however, the width of the black scaling bar changes each year, sometimes substantially, because the total number of prey detected each year ($N$) varied. (a) 2009, $N = 223$, (b) 2010, $N = 266$, (c) 2011, $N = 663$ and (d) 2012, $N = 336$.

~70% of the value for the summary web. Lower values reflect the fact that subsets of the full data matrix were selected for bootstrapping [40].

## Interaction evenness

The broad overlap in confidence intervals around median values of interaction evenness suggests no major changes in this index across years (Fig 5A) nor seasons (Fig 5B). Median values for years and seasons were nearly identical to the value for the summary web. Interaction evenness of the summary web was 0.93, which was calculated for detected trophic interactions. This high value appears to be consistent with the number, widths, and distribution of trophic interactions in the graphical depiction of summary web structure (Fig 1).

## % IGPrey

Most interactions were part of IGP modules. In the summary web, 99% of spider-spider predation events involved an IGPredator and an IGPrey. The exception was six individual feeding events involving Corinnidae, for which predation on Pisauridae (3/125) and Hahniidae (3/125) was not part of an IGP module because these three spider families had no non-spider prey in common. Among non-spider prey, 97% of detected predation events were part of an IGP module. Considering pathways alone, i.e., ignoring number of predation events, IGP modules dominated (>90%). From the top-down perspective, 37 predation pathways were part of an IGP module, only two were not. Only four modules were asymmetrical (non-reciprocal) IGP. Among the 41 types of prey found in spider guts, only three were not part of an IGP module. Further details appear in S3 Table.

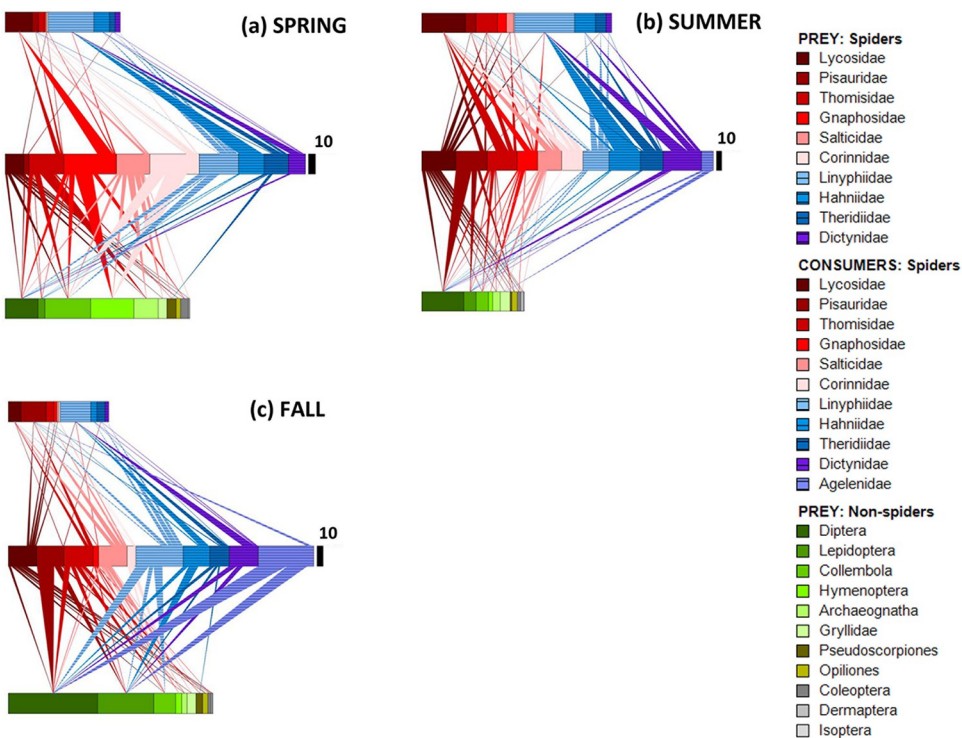

**Fig 3. Food webs for spring, summer and fall (years combined).** Key as in Fig 2. The width of the black scaling bar changes because the total number of prey detected each season (*N*) varied. (a) Spring, *N* = 421, (b) Summer, *N* = 575 and (c) Fall, *N* = 492.

In the summary web, % IGPrey in spider diets was close to 50%, varying little across the four years (seasons pooled) (Fig 6A). In contrast, the frequency of predation on IGPrey varied seasonally (years pooled) (Fig 6B). In the summer, % IGPrey was 65% ± 4%, about 1.8x higher than in spring and fall (38% ± 5% and 33% ± 4%, respectively). The pattern of this seasonal shift in % IGPrey was similar across the four years of the study (Fig 7).

Most spider-spider predation (~90%) was confined to families within the same foraging mode. Web spinners accounted for only ~7% of cursorial IGPrey, and ~11% of web-spinner IGPrey were cursorial spiders.

## Discussion

To advance ecological theory on food web structure and dynamics, mathematical models should be tested against the behavior of natural food webs to validate model predictions. Most empirical food webs constructed to date assume an equilibrium state and are summary webs derived from pooled seasonal or yearly data, ignoring temporal constancy or variability in food web structure [2]. Furthermore, changes in the pattern of IGP in quantitative food webs rarely have been tested, despite the implications of extensive IGP for the dynamics of natural food webs. This study addressed these knowledge gaps by investigating the structure of a spider-focused food web on a deciduous forest floor during spring, summer, and fall over four years using PCR to detect prey DNA in 3,300 adult spiders. The structure of this spider-focused food web was remarkably constant over four consecutive years. Thus, models designed to explore the dynamics of this food web should incorporate the consistent behavior of three indices of food web structure (restricted connectance, interaction evenness, and % IGPrey) when the web is not subject to external perturbations.

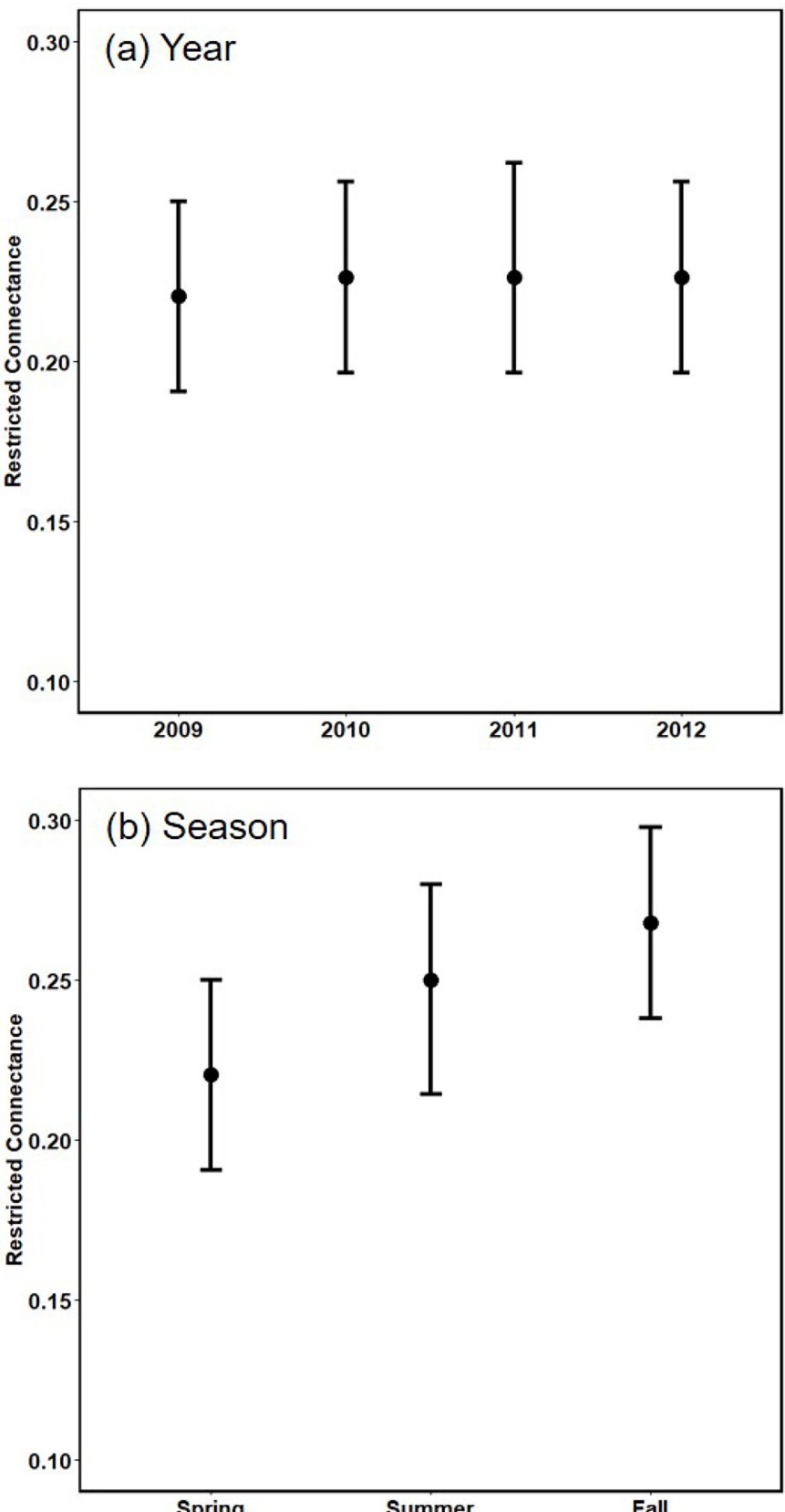

**Fig 4. Median restricted connectance, with 95% BCI.** (a) Year (seasons pooled) and (b) Season (years pooled). Median values are about ~25% of the maximum possible.

Below we explore some possible causes, and implications, of yearly constancy in these three indices, followed by suggestions on exploiting this spider-focused food web to develop theories of food web dynamics with many generalist predators and high degrees of IGP.

## Restricted connectance

Restricted connectance, which is the proportion of detectable, biologically feasible linkages that are realized, can be used to understand how nodes are connected in a food web. We found high temporal constancy in restricted connectance in a spider-focused food web, our least surprising result. One would predict minimal variability for a binary (presence/absence) index when all consumer and prey nodes are broadly defined taxonomically. Just one detectable connection between individual taxa within a pair of consumer and prey nodes contributes the same to a binary index as do numerous between-taxa connections. This redundancy explains the temporal constancy of the connectance index for our food web.

The index of restricted connectance (0.36) in this web is higher than typical connectance measures, which are calculated for community-level food webs in which non-likely trophic connections are included. Values of these indices range from ~ 0.03 to ~ 0.3, with an average around 0.1 [41]. A focus on spiders and their likely non-spider prey created a restricted sink web, which is the primary reason that our modified connectance index is higher than standard measures.

## Interaction evenness

Unlike connectance, interaction evenness is a quantitative measure whose value will change if the relative frequencies of trophic interactions change even when the binary index of connectance remains constant. Thus, one would predict that interaction evenness would be the more variable index. But such was not the case for this web. Interaction evenness exhibited no greater seasonal and yearly variability than did restricted connectance. Temporal constancy of interaction evenness likely resulted from the nature of the nodes in a spider-focused web. All nodes in this food web are broadly defined taxonomically; furthermore, the consumers are generalists. In fact, most spider families in this web preyed across several of the broadly defined prey nodes. The ability of these highly generalist predators to switch prey species in response to changes in their availability, and high redundancy in suitable prey (non-spider or spider) within a node, likely dampened effects on interaction evenness of temporal variation in particular species. Of course, it is also possible that densities of all node components were similar across years; we have no data on densities to evaluate this possibility. Nevertheless, densities of many species in all nodes certainly varied seasonally, yet interaction evenness exhibited no clear seasonal variation.

## % IGPrey

Three findings stand out. First, most interaction pathways existed within IGP modules (> 90%), most of which were reciprocal IGP (S3 Table). In the summary web, most non-spider prey were a shared resource in an IGP module (97%); and practically all spider-spider predation events (99%) involved consumption of an IGPrey.

Secondly, the overall rate of predation on other spiders, pooled across seasons, was remarkably similar across years and was relatively high (~50%). This rate is at least 3-4x higher than

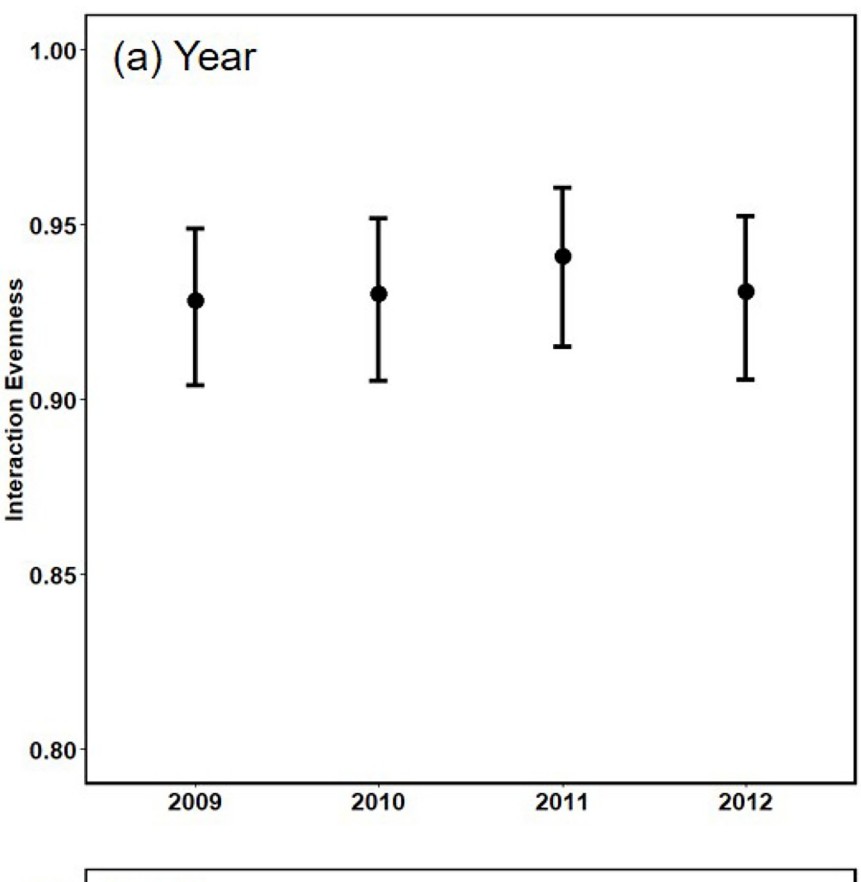

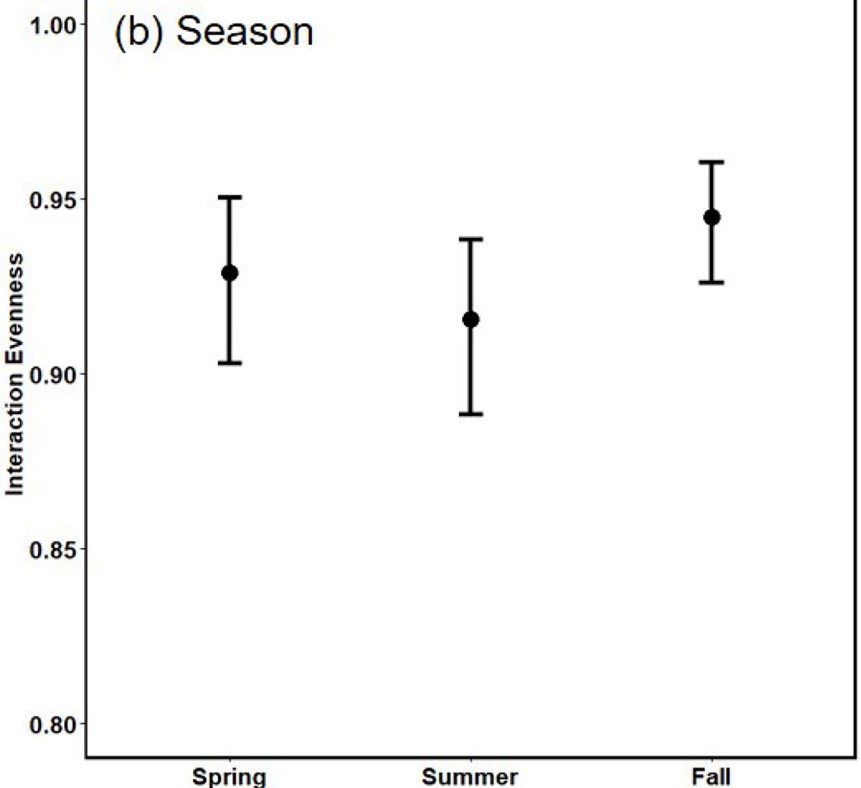

**Fig 5. Median interaction evenness, with 95% BCI.** (a) Year (seasons pooled) and (b) Season (years pooled). Median values are about ~90% of the maximum possible.

rates of predation between spider families in agricultural systems as revealed by molecular techniques [42–45]. However, comparing these rates with a forest-floor system is problematic because of differences in habitat structure and the invertebrate fauna [46]. There appear to be no other studies in forests that use molecular techniques to measure either rates of predation on spider IGPrey or even simple rates of predation between spider families.

A third major finding is that the proportion of IGPrey in spider diets varied seasonally in a pattern that was repeated over four years. Summer rates were consistently ~1.8x higher than rates in spring or fall, possibly due to lower densities of alternative, non-spider prey because of warmer, drier conditions during summer months. Higher summer percentages of IGPrey in spider diets might also reflect seasonal variation in spider reproduction. Male spiders actively seek females; if most species matured in summer, increased activity by recently matured males could have contributed to higher rates of predation on spiders by other spider families in summer. Seasonal shifts in % IGPrey also might have reflected changing ratios of the number of adult spiders sampled for prey DNA to numbers of juveniles of other spider families (potential prey). We have no data to test the relative contribution of these hypotheses for our system. The underlying causes of higher values of % IGPrey in summer merit further research.

Thus, a very high fraction of all interaction pathways existed within IGP modules, most of which exhibited reciprocal IGP; rates of spider predation on IGPrey were relatively high and stable across years (seasons pooled); and % IGPrey in spider diets exhibited a repeatable seasonal pattern each year. These food web motifs should be incorporated into models of trophic interactions among spiders and their arthropod prey in temperate forest-floor systems.

## Pathways of intraguild predation

Another clear pattern in this spider-focused food web is the concentration of spider-spider predatory interactions to families grouped by foraging mode: cursorial spiders and web spinners. The consistent grouping of IGP interactions by foraging mode over four years strongly suggests that cursorial spiders and web spinners comprise two recognizable channels of indirect effects in the forest-floor food web. These channels appear only weakly connected via IGPrey, as just ~10% of direct spider-spider interactions involved spiders from the other channel. This pattern appears to contrast with a recent meta-analysis that documented much-higher rates (sometimes > 50%) of reciprocal predation between cursorial and web-spinning spiders [47]. This possible inconsistency merits some explanation.

Michalko et al. [47] analyzed data from 67 studies that included observations of spider-spider predation in the field and feeding studies in the laboratory. They analyzed 1078 prey records for 72 samples of consumers (5–71 prey records per sample) from three strata (one of which was "ground") in forests, agroecosystems, and semi-natural systems. When 1078 prey records are divided across nine habitat categories, data for the ground layer of forests is not sufficient to compare with our 3300 prey records for this habitat type. Nevertheless, it is striking that web spinners constituted only ~10% of the spider prey for "hunters" (roughly equivalent to our "cursorial" category) in the ground layer of semi-natural habitats. Data were insufficient to categorize spider prey for web spinners of the ground layer. It is difficult to directly observe the prey of spiders in the litter layer of forests, so it is not surprising that such data were incomplete in the meta-analysis of Michalko et al. [47]. Our data set provides valuable data points for future meta-analyses similar to theirs.

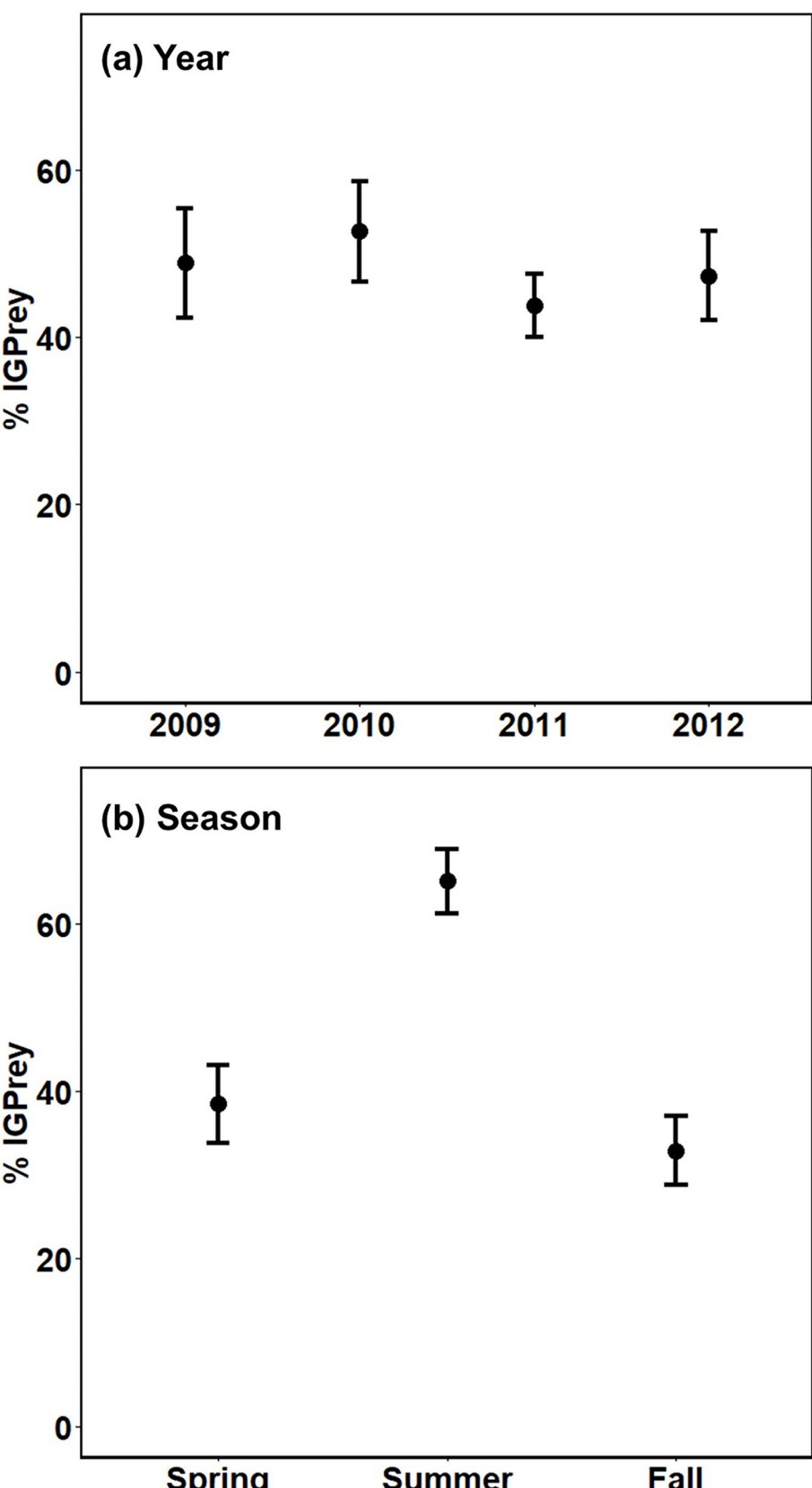

**Fig 6. % IGPrey, with 95% CI.** (a) Year (seasons pooled) and (b) Season (years pooled). % IGPrey is the percentage of total prey that is other spiders that are part of an IGP module.

The segregation of spider-spider predation into two channels in forests is likely more pronounced than our results suggest. Non-sexual cannibalism occurs widely among all instars in numerous spider species [48]. Although cannibalism is more frequent between stages that differ substantially in size, cannibalism among recently hatched spiderlings can be a major mortality factor for lycosids on the forest floor [49]. Intra-family predation may even be more prevalent than cannibalism, because spiders within the same genus, and often in different genera within the same family, exhibit similar prey-foraging behaviors. We could not detect within-family predation because we did not use genus-level primers, and using standard DNA techniques to detect cannibalism in natural populations is not feasible because prey and predator DNA sequences are too similar. The likely widespread occurrence of cannibalism and intra-family predation makes it is reasonable to hypothesize that much more than 90% of spider-spider predation on the forest floor is restricted to one of two interaction channels defined by foraging mode.

Although this pattern supports the hypothesis that susceptibility to attack by other spiders relates directly to behaviors that are critical to foraging for prey, other factors must play a role. Web spinners should be exposed to predation when searching for new web sites, and when searching for mates–why, then, do they not appear more often in the diets of cursorial spiders? Species of several web-spinning families of the ground layer also forage for prey off the web [16] —should not that behavior also expose them to predation by cursorial families? Cursorial spiders may be able to avoid webs, but why do they not fall prey more frequently to those web spinners that also forage off the web? Coupled with these questions is the finding that predation between web spinners and cursorial spiders may be more frequent in habitats other than the forest floor [47]. Furthermore, if the gut DNA of juvenile spiders on the forest floor were to be analyzed, inter-channel predation might turn out to occur more—or maybe less—often than our focus on adult predators revealed. Clarifying the boundaries of these two interaction channels awaits more analyses of spider prey at the species level for a wider spectrum of spider life stages. Recently implemented NGS barcoding techniques with spiders [44, 45, 50–52] promise major insights if applied to these two interaction channels in spider-focused food webs.

## Implications for modeling the dynamics of food webs with widespread IGP

The long debate over the implications of extensive IGP for the dynamics and stability of terrestrial food webs has included feedback between theoretical and empirical research. A major contribution to resolving this debate, and related ongoing debates over food web ecology, could come from detailed, long-term research on model natural systems [53]. The spider-focused web of the forest floor is an excellent candidate for a suitable empirical model. The first, and obvious, advantage is the prevalence of IGP modules with high rates of reciprocal predation between spider families.

The second advantage is the temporal constancy in the pattern of IGP across years. A recent 10-yr study found a correlation between changes in trophic structure and changes in population dynamics in a tri-trophic parasitic system [54]. Such a result offers hope that an empirical food-web model apparently at equilibrium, at least for a particular level of node definition, could be an appropriate tool for testing mathematical models assuming equilibrium. Of course, constancy in structure does not necessarily imply absence of counterbalancing changes in the dynamics of these interaction pathways. Similarly, temporal constancy does not imply an equilibrium state resistant to perturbations. An advantage of this spider-focused food web as an empirical model is

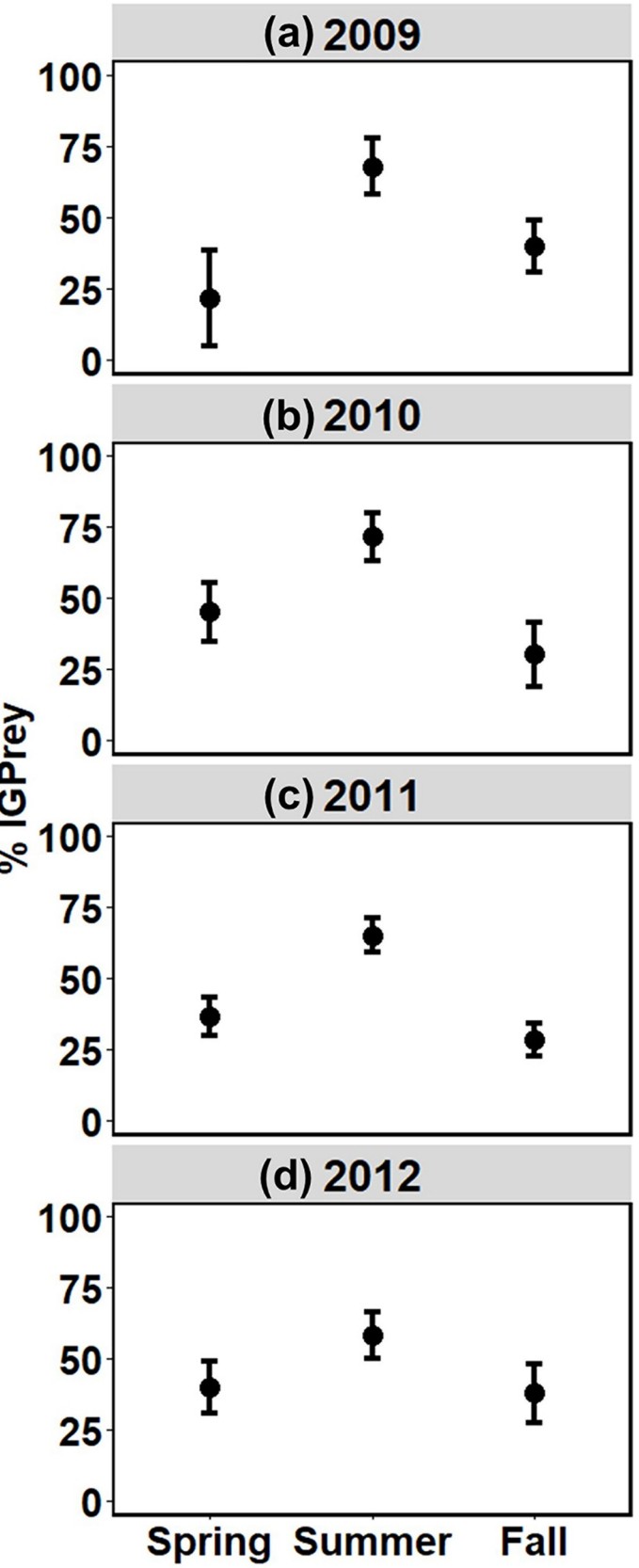

**Fig 7. Pattern of seasonal change in % IGPrey, with 95% CI, across four years.** (a) 2009, (b) 2010, (c) 2011 and (d) 2012. The seasonal shift in % IGPrey was not paralleled by any seasonal change in the proportion of interaction pathways that were IGP modules (S3 Table). The best model for the 4 x 3 x 2 contingency table of changes in % IGPrey did not include an interaction between season and year; thus, there is no evidence suggesting that the pattern of seasonal change in % IGPrey differed across years. The best model did include an effect of season, but no effect of year, on % IGPrey–a result that is consistent with the pattern here and in Fig 6. Details of the modeling results can be accessed on Dryad: https://doi.org/10.5061/dryad.dz08kps43.

that mechanisms underlying the stable seasonal pattern of IGP–and the dynamic implications of this pattern—can be revealed by perturbing the system in field experiments.

In fact, the major advantage of the proposed empirical model may be its proven suitability for controlled, replicated field experiments. The arthropod food web of leaf litter of the forest floor has proven amenable to manipulations of the detrital resource base, long-term rainfall, spider densities, and densities of other generalist arthropod predators (for recent examples and discussions of earlier related studies, refer to [13, 55–59]). Long-term perturbation experiments will be especially valuable in testing theory when coupled with monitoring of changes in feeding connections between food web nodes across different time scales using rapidly evolving DNA, fatty acid, and stable isotopic techniques [7, 60].

## Conclusions

We discovered several remarkably distinct attributes of IGP in a spider-focused forest-floor food web. Almost all interactions occurred within IGP modules. Yearly rates of predation by spiders on IGPrey were high and did not vary substantially over four years. Rates were higher in summer than spring or fall, and this seasonal pattern itself was consistent over time. Finally, IGP occurred predominantly within two distinct pathways broadly defined by spider foraging mode. Researching the causes and implications of these temporally constant patterns of IGP would contribute to a deeper understanding of the dynamics of species-rich terrestrial food webs with numerous generalist predators.

Furthermore, because the spider-focused food web is widespread throughout terrestrial ecosystems, our results highlight the need to consider the dynamics of IGP in studying terrestrial ecological networks. As knowledge of broadly defined ecological networks is foundational to understanding how ecosystems are structured [61, 62], future research on networks of interactions in terrestrial systems that incorporate patterns of IGP will provide a more complete picture of how terrestrial ecosystems may respond to environmental perturbations, particularly in the era of global change.

## Supporting information

**S1 File. Rates of prey detection.**
(DOCX)

**S1 Table. Number of spiders per genus analyzed, and testing positive, for prey DNA.**
(PDF)

**S2 Table. A more detailed analysis of patterns in the summary web.**
(PDF)

**S3 Table. Comparison of seasonal shifts in % IGPrey and seasonal shifts in interaction pathways.**
(PDF)

## Acknowledgments

We thank the Forest Preserve District of Cook County for permission to collect spiders from the Swallow Cliff Woods, and Dr. Petra Sierwald and her colleagues at The Field Museum for assistance with spider identification. We thank the following undergraduates for invaluable assistance in the field and laboratory: Melissa Mariscal, Erin Barry, Shefali Batra, Stephanie Mendoza, Sarah Vega, Yuliya Voskobiynyk, Lan Dam, Gediminas Gaidamavicius, Denise Hernandez, Shaheera Fatima, Sukhvinder Chada, Khris Villarin, Brian Sebeh, Canisha Howard, Diana Gorczak, Aubrey Jones, Kristine Lindsey, Irish Quindara, Sean Arca, Meet Shah, Saager Patel, Chantell Urquiza, Sagar Anantatmula, Safia Zidan, Francoise Kaleta, Halema Zayyad, Francisco Casambre and Anne Azzo.

## Author Contributions

**Conceptualization:** David H. Wise, Robin M. Mores.

**Data curation:** Robin M. Mores.

**Formal analysis:** David H. Wise, Jennifer M. Pajda-De La O, Matthew A. McCary.

**Methodology:** Robin M. Mores, Jennifer M. Pajda-De La O.

**Visualization:** David H. Wise.

**Writing – original draft:** David H. Wise, Matthew A. McCary.

**Writing – review & editing:** David H. Wise, Robin M. Mores, Jennifer M. Pajda-De La O, Matthew A. McCary.

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
