## [Decision Letter · Decision Letter 0]

26 Jul 2023

PONE-D-23-17199Temporal constancy in the structure of a spider-focused food web with high rates of intraguild predationPLOS ONE

Dear Dr. McCary,

Thank you for submitting your manuscript to PLOS ONE. After careful consideration, we feel that it has merit but does not fully meet PLOS ONE’s publication criteria as it currently stands. Therefore, we invite you to submit a revised version of the manuscript that addresses the points raised during the review process.

We look forward to receiving your revised manuscript.

Kind regards,

Kleber Del-Claro, PhD

Academic Editor

PLOS ONE

Additional Editor Comments:

This is a very interesting study, well-done and well-written, which will make a valuable contribution to a better comprehension of spider holes in food webs. Both reviewers provided positive comments and suggested minor modifications. I recommend that the authors address aspects of the theory of ecological networks in the discussion, as it could make the paper more appealing to a wider audience. There are recent books on Plant-animal interactions and Ecological Networks that present interesting chapters that have built this bridge. Finally, congratulations on the excellent work.

Reviewers' comments:

Reviewer's Responses to Questions

**Comments to the Author**

1. Is the manuscript technically sound, and do the data support the conclusions?

Reviewer #1: Yes

Reviewer #2: Yes

2. Has the statistical analysis been performed appropriately and rigorously? 

Reviewer #1: Yes

Reviewer #2: Yes

3. Have the authors made all data underlying the findings in their manuscript fully available?

Reviewer #1: Yes

Reviewer #2: Yes

4. Is the manuscript presented in an intelligible fashion and written in standard English?

Reviewer #1: Yes

Reviewer #2: Yes

5. Review Comments to the Author

Reviewer #1: Temporal constancy in the structure of a spider-focused food web with high rates of intraguild predation

The study´s goal was to “assess temporal variation in food web structure”. Using gut content of spiders they have collected themselves, the authors found that almost all interaction occurred within the same category (ie, web builders or cursorial spiders), that predation rates were seasonal and not much variation throughout the years. The study was well conducted, data extensively collected and well analyzed. The paper is also clear, but I do have minor suggestions that should be addressed. I also do not think there is a match between the title and main goal (these two match) and main conclusions, which go beyond the title and main goal.

52 Please improve the link of this second paragraph with the first one

100 “reasonable to expect that both connectance and interaction evenness would exhibit

considerable constancy.”

Males and females typically differ in activity because the former wander looking for the latter. Wandering animals are more likely to fall prey of sit and wait predators such as spiders. Because reproduction is seasonal, the authors could elaborate, in the discussion section, how that influences the temporal stability of the indices.

143 Because pisaurids were sampled, I imagine there was a river crossing the sampled area. How was this habitat heterogeneity considered when sampling? Were samplings homogeneous throughout the years with respect to microhabitats and specific areas? The different number of sampling days was corrected when analyzing data?

167 Could you please elaborate on pisaurids? Fig 1 shows they consumed spiders of a couple of families

and lepidopterans and flies, but pisaurids are semi-aquatic spiders. Is there natural history data that would allow discussing in which context these predations events happened?

172-174. I might have missed it earlier, but here is where I first found an explanation on why non-spider species were included in the analysis, Because the whole theoretical framework of the paper was about IGP and most of the non-spider species cannot prey on spiders (and therefore are also “non-IGPs”), I think the authors could better explain their inclusion in the dataset.

177 Aracheognatha typo

226 - Only 17 out of 3300 spiders tested positive for two different prey items;

Could please elaborate on that? Why does that happen when we know such generalist predators feed upon several species? Is it because the method usually detects very recent prey? Whatever the explanation is, it would help understanding such apparent unusual and unexpected result.

276 we could not test for intra-family predation

Are there any available data to give an idea of how important this is and how the absence of it influences the conclusions? I would assume there is a lot of intra-family predation

499-502 And can we learn something from this comparison?

506-513 What do we know about seasonal variation in populations of these studied species, including non-spider ones? Their relative abundances certainly influence all the studied interactions and may help explain shifts in IGPrey.

520-542 This section compares the results with Michalko´s paper, which I agree is important. However, I was expecting hypotheses to explain why the consistent grouping of IGP interactions by foraging mode (web vs cursorial). Is it possible to suggest reasons for this pattern?

Reviewer #2: ---

Dear Editor,

This study focused on the spider-focused food web found on the forest floor and aimed to understand the temporal variation in this web, which is characterized by widespread IGP.

The researchers collected 3,300 adult spiders from a deciduous forest floor during spring, summer, and fall over four years. They used multiplex PCR to detect prey DNA in the spiders' guts to identify the prey consumed by the spiders. The web was found to be tripartite, consisting of 11 consumer nodes (spider families) and 22 resource nodes (11 non-spider arthropod taxa and the 11 spider families).

The study found that most of the spider-spider predation involved IGPrey, and about 90% of these interactions occurred between spider families within the same broadly defined foraging mode, such as cursorial spiders or web spinners.

The researchers analyzed the temporal constancy of the web structure using three indices: restricted connectance, interaction evenness, and % IGPrey. The results showed that the web structure, as indicated by the first two indices, remained relatively constant across years and seasons, with overlapping confidence intervals.

However, the % IGPrey varied seasonally, with the highest rate observed in summer than in spring and fall. This seasonal pattern was consistent across the years of the study.

The study highlights the importance of considering extensive spider predation on IGPrey and its consistent seasonal variation in frequency within specific interaction pathways when modeling the dynamics of forest-floor food webs.

Overall, the manuscript is well written. The objectives were well presented. Methodological procedures were adequately described in most parts of the text.

The relationships among spiders within the guild are very intriguing and help to understand the complexity of interspecific relationships among spiders and the role of these relationships in shaping and maintaining food webs. The findings from this study may contribute to future studies of arthropod community maintenance associated with leaf litter.

In my opinion, the study brings very important innovations to consumer relationships in spider communities and will appeal to a significant portion of PlosOne's readership. Although I believe the manuscript is of high quality, I still have some questions about the data collection.

-How many samples were collected per day at each location?

-The study was conducted in a 320 ha preserve. How was the site selected for leaf litter collection and sorting?

-For example, was the leaf litter collected in 50cm x 50cm plots?

-Is it possible to estimate the total area that was sampled based on the total size of the litter collection plots?

-The sampling was done between 10:00 and 16:00. Can I assume that the spiders forage more at night? If so, could we expect variation in spider abundance or variation in guild or family composition?

-It would be interesting to see images of the study area and perhaps a figure or map showing the shape of the reserve and the distribution of leaf litter collection sites in the area.

I emphasize again that the study is very exciting and thank you for the opportunity.

6. PLOS authors have the option to publish the peer review history of their article (what does this mean?). If published, this will include your full peer review and any attached files.

Reviewer #1: No

Reviewer #2: No

---

## [Author Response · Author response to Decision Letter 0]

9 Sep 2023

RESPONSES TO EDITOR and REVIEWERS, PONE-D-23-17199

Editor

1. This is a very interesting study, well-done and well-written, which will make a valuable contribution to a better comprehension of spider roles in food webs. Both reviewers provided positive comments and suggested minor modifications. I recommend that the authors address aspects of the theory of ecological networks in the discussion, as it could make the paper more appealing to a wider audience. There are recent books on Plant-animal interactions and Ecological Networks that present interesting chapters that have built this bridge. Finally, congratulations on the excellent work.

We are encouraged by this positive comment, as a lot of work went into this research. The two reviewers’ comments made the manuscript clearer and improved its overall quality. We also appreciate the suggestion to place our research in the broader context of ecological networks. While we agree that this topic is broadly relevant to our research, the suggestions from Reviewers 1 and 2 regarding the need for more details on the natural history of spiders, our collection methods, and our definition of the prevalence of intraguild predation in a consumer-focused food web, rendered it difficult to address how our research on a spider-focused food web relates directly to theories of ecological networks without sounding off-topic or going beyond the data. However, we added a few sentences to the “Conclusions” paragraph of the Discussion that emphasize how our research can advance understanding of complex ecological networks that include intraguild predation:

“Furthermore, because the spider-focused food web is widespread throughout terrestrial ecosystems, our results highlight the need to consider the dynamics of IGP in studying terrestrial ecological networks. As knowledge of broadly defined ecological networks is foundational to understanding how ecosystems are structured (61, 62), future research on networks of interactions in terrestrial systems that incorporate patterns of IGP will provide a more complete picture of how terrestrial ecosystems may respond to environmental perturbations, particularly in the era of global change.”

Reviewer #1

2. I also do not think there is a match between the title and main goal (these two match) and main conclusions, which go beyond the title and main goal

Excellent point. Our new title is more informative: 

“Pattern of seasonal variation in rates of predation between spider families is temporally stable in a food web with widespread intraguild predation.” 

3. 52 Please improve the link of this second paragraph with the first one

Another excellent point. These two new sentences appear at the end of the first paragraph, providing a direct link to the topic sentence of the second paragraph: 

“Identifying empirical food webs that appear to be at equilibrium over several years is critical to testing theories about factors that affect food web stability. One factor hypothesized to influence food web stability is the prevalence of trophic-level omnivory.”

4. 100 “reasonable to expect that both connectance and interaction evenness would exhibit considerable constancy.” 

Males and females typically differ in activity because the former wander looking for the latter. Wandering animals are more likely to fall prey of sit and wait predators such as spiders. Because reproduction is seasonal, the authors could elaborate, in the discussion section, how that influences the temporal stability of the indices.

We do not agree that seasonal spider reproduction would likely have a discernable effect on patterns of connectance and interaction evenness, which reflect interactions with both spider and non-spider prey. The reviewer may have been thinking about how seasonal changes in the activity of males could drive some of the seasonal variation in percentages of IGPrey (our third index of food web structure) in the diets of spiders. This is an excellent point, which prompted us to add the following sentences to the section “% IGPrey” in the Discussion: 

“Higher summer percentages of IGPrey in spider diets might also reflect seasonal variation in spider reproduction. Male spiders actively seek females; if most species matured in summer, increased activity by recently matured males could have contributed to higher rates of predation on spiders by other spider families in summer.”

5. 143 Because pisaurids were sampled, I imagine there was a river crossing the sampled area. How was this habitat heterogeneity considered when sampling? Were samplings homogeneous throughout the years with respect to microhabitats and specific areas? The different number of sampling days was corrected when analyzing data?

There was no river crossing the collecting locations, but there were some wet areas. Please see the comment below relating to pisaurids.

Our collecting procedures were not intended to be quantitative; thus, we gathered no data on densities of spiders or non-spider prey, nor did we block out regions of the forest within which our collecting areas were located. We aimed to reduce possible biases due to forest-floor heterogeneity by searching relatively large areas on each collecting day and by having many collecting days (93) in different locations. Sampling of spiders was not spatially stratified over time (i.e., not strictly “homogeneous throughout the years with respect to microhabitats and specific areas”). As explained in the revised “Materials and methods” section, we did examine all types of microhabitats on each collecting day. In this sense, sampling was somewhat homogeneous over years. Sampling of spider densities was not necessary for our research goals, and there is no meaningful way to “correct” for number of days that spiders were collected. 

The above questions made us realize that our description of the protocol for collecting spiders was inadequate. We have clarified the procedures in the “Sampling of spiders and non-spider prey” sub-section of Methods and Materials by revising the section sub-title; and by revising the first two, and last, paragraphs, which now read as follows:

“Collecting spiders and non-spider prey

Our goal was to search the ground layer and low understory as thoroughly and systematically as possible, so that we would collect enough spiders from less-abundant families to yield the same number of spiders per family analyzed for prey DNA. We did not estimate spider densities. All collections were made between 1000 and 1600 hours. We collected from a different location each day. The size of the area searched each day was not measured and varied with the number of searchers. Collecting areas were widely distributed throughout Swallow Cliff Woods, but we did not subdivide the Woods into sampling regions. Most terrain was upland forest, but some collections were taken from a few scattered wet/marshy areas. The number of collecting days in each season was spring (31), summer (33), and fall (29) over the years 2009, 2010, 2011 and 2012; the number of days per year was 33, 12, 34 and 14, respectively.

On each collecting day, we used both litter sifting and simple searching to capture spiders from several microhabitats. For litter sifting, we placed litter collected by hand into a flat tray (58 cm x 17 cm x 15 cm) with a screen bottom. This tray was shaken over a second tray of the same size with a solid bottom, allowing arthropods to fall through the screen to be collected by hand or aspirator. Sifted litter was returned to its original location. Spiders were also collected by hand from the litter surface, open areas in the litter, logs, low vegetation up to ~1m, and tree trunks up to ~2m. Individual spiders were placed in separate labelled vials.

. . . . . . . . . 

. . . . . . . . . 

 Non-spider arthropod prey were also collected for primer development. They were not sampled quantitatively, but were simply selected due to their apparent abundance in leaf litter and/or activity just above the litter layer, and their likely occurrence in the diets of at least one spider family. . . . .”

6. 167 Could you please elaborate on pisaurids? Fig 1 shows they consumed spiders of a couple of families and lepidopterans and flies, but pisaurids are semi-aquatic spiders. Is there natural history data that would allow discussing in which context these predations events happened?

Two-thirds of the pisaurids analyzed were from the genus Pisaurina (S1 Table). This genus is not semi-aquatic but is found in vegetation (grasses and bushes) in meadows and forests, sometimes on tree trunks. The other pisaurid genus we analyzed, Dolomedes, is semi-aquatic (it feeds on aquatic insects (and fish on occasion!) but probably also on terrestrial prey along the shore). In our submitted manuscript we neglected to point out that we collected spiders from some wet/marshy areas (now included in the first paragraph of “Materials and methods”; see response to Comment #5). We decided not to clarify why we collected pisaurids because we tried to minimize natural history details to keep the manuscript focused on general concepts. Arachnologists reading the paper who consult S1 Table and read the revised description of the collecting areas will readily interpret the pattern for pisaurids.

7. 172-174. I might have missed it earlier, but here is where I first found an explanation on why non-spider species were included in the analysis, Because the whole theoretical framework of the paper was about IGP and most of the non-spider species cannot prey on spiders (and therefore are also “non-IGPs”), I think the authors could better explain their inclusion in the dataset.

We understand why the reviewer might be confused on this point, because spider-spider predation (and indeed, predation between any two generalist predators) sometimes is loosely defined as IGP, which is incorrect because such an interaction constitutes IGP only if the two spiders (or any two generalist predators) share a resource on a lower trophic level. In our spider-focused tripartite web, the shared resource was a non-spider taxon (which could be a predator, but usually was not). Thus, not all spider-spider predation is necessarily IGP; in fact, we uncovered some (but only a few) three-level trophic chains that were “Spider 1 – Spider 2 – Non-shared non-spider resource.” Although almost all spider-spider predation in our web was on spider IGPrey, we defined the prevalence of IGP to be % IGPrey in spider diets to separate spider-spider predation events that were not part of an IGP module. Because many readers may also use the less-precise definition of IGP (i.e., identifying all spider-spider predation as IGP), and some may be unfamiliar with the “focused-web” approach to investigating food-web structure (i.e., the “sink” and “source” webs of the classical food web literature), we have modified the following sections to put our usage in sharper focus:

Abstract: 

We have expanded the abstract (additions in bold) and made minor deletions to keep within the 250-word limit:

“Intraguild predation (IGP) – predation between generalist predators (IGPredator and IGPrey) that potentially compete for a shared prey resource – is a common interaction module in terrestrial food webs. Understanding temporal variation in webs with widespread IGP is relevant to testing food web theory. We investigated temporal constancy in the structure of such a system: the spider-focused food web of the forest floor. Multiplex PCR was used to detect prey DNA in 3,300 adult spiders collected from the floor of a deciduous forest during spring, summer, and fall over four years. Because only spiders were defined as consumers, the web was tripartite, with 11 consumer nodes (spider families) and 22 resource nodes: 11 non-spider arthropod taxa (order- or family-level) and the 11 spider families. Most (99%) spider-spider predation was on spider IGPrey, and ~90% of these interactions were restricted to spider families within the same broadly defined foraging mode (cursorial or web-spinning spiders). Bootstrapped-derived confidence intervals (BCI’s) for two indices of web structure, restricted connectance and interaction evenness, overlapped broadly across years and seasons. A third index, % IGPrey (% IGPrey among all prey of spiders), was similar across years (~50%) but varied seasonally, with a summer rate (65%) ~1.8x higher than spring and fall. This seasonal pattern was consistent across years. Our results suggest that extensive spider predation on spider IGPrey that exhibits consistent seasonal variation in frequency, and that occurs primarily within two broadly defined spider-spider interaction pathways, must be incorporated into models of the dynamics of forest-floor food webs.” 

Introduction: We have made four additions:

 Second paragraph:

 “ . . . . . . The simplest example of trophic-level omnivory is intraguild predation (IGP) (4), defined most broadly as a module (5) in which two generalist predators that share a potentially limiting resource on a lower trophic level (a herbivore, microbivore, detritivore or another predator) also feed upon each other. . . . . . . ”

 Third paragraph:

“. . . . Changes in our third index of web structure, “% IGPrey” (% of all prey in the focal consumer that are IGPrey), could reflect changes in relative energy flow through the two channels of IGP modules. . . . .”

 Fifth paragraph:

 “Because only spiders were defined as consumers in our spider-focused food web, it is a tripartite web with 11 consumer nodes (spider families) and 22 resource (prey) nodes: the 11 spider families and 11 non-spider arthropod taxa defined at a broad scale (order or family). . . . . . .”

 Addition of a new paragraph after the sixth paragraph:

 “Although predation between two generalist predators sometimes is described loosely as “intraguild predation,” we use the strict definition of IGP and do not describe all spider-spider predation as examples of IGP. In our tripartite web, predation between two spider families was IGP only if the two families shared a non-spider prey taxon. Thus, to exclude spider-spider predation events that were not part of an IGP module, we defined the prevalence of IGP as the % of all prey that were IGPrey in spider diets (i.e., % IGPrey).” 

8. 177 Aracheognatha typo

Corrected; now Archaeognatha

9. 226 – “Only 17 out of 3300 spiders tested positive for two different prey items”;

Could please elaborate on that? Why does that happen when we know such generalist predators feed upon several species? Is it because the method usually detects very recent prey? Whatever the explanation is, it would help understanding such apparent unusual and unexpected result.

We agree that this unexpectedly low rate of two positives per spider deserves elaboration, especially because the result is not so surprising once possible causes are evaluated. We now refer to a new supplemental file, “S1 File”, which gives several possible explanations for the low rate of detection of multiple prey items. We have placed this material in a supplemental file because inserting this technical discussion in the manuscript would disrupt the flow of the narrative.

Addition to last paragraph of “Spider gut-content testing” in “Materials and methods”:

“. . . . . . . Only one spider tested positive for more than two prey: a thomisid tested positive for 7 different prey taxa, which was considered an outlier and was removed from the analysis (refer to S1 File for a discussion of possible causes of the low number of spiders testing positive for more than one prey item). . . . . .” 

New supplement: S1 File:

“S1 File: Rates of prey detection

Of 3299 spiders analyzed (3300 minus the possibly anomalous thomisid), 1,467 tested positive for one prey item, 17 tested positive for two prey taxa, and none tested positive for ≥ 3 prey. If the probabilities of detecting 0, 1, 2, 3 . . . prey taxa in one spider are independent, one would expect 7.5% of the spiders analyzed to test positive for two or more prey (fitting a Poisson process with λ = 0.45 to our data). In our data set, only 17 (0.5%), tested positive for ≥ 2 prey. Given our large sample size, prey detection clearly was not a simple Poisson process. What might be causing this deviation from independence i.e., why does the probability of detecting DNA of a second prey taxon depend upon the presence of another taxon in the spider gut? Several possible explanations come to mind that are related to spider ecology, our collecting protocol, and possible limitations of multiplex PCR. The most likely explanations are the following:

• Spider populations often are food limited and direct observations have revealed that spiders often may consume only one prey item per day (1).

• Adult males in many spider families do not feed as frequently as adult females (2). Over half (55%; S1 Table) of spiders we analyzed were adult males. 

------The above two factors suggest that many spiders likely had eaten only one prey item in the previous day or two. A higher rate of predation could have increased the chances of detecting multiple types of prey DNA before it had broken down into fragments too small to be detected by the primers utilized in our multiplex PCR analysis (King et al. 2008). 

(Note: one might argue that the contribution of the above two factors is directly reflected by the number of “failures” (spiders in which prey was not detected) used to fit the Poisson distribution to the observed data. If we were to accept this argument, we would still need to explain why multiplex PCR failed to detect the expected number of multiple predation events.) 

• The 93 collecting days were distributed across three seasons in each of four years. Not all potential prey would have been consistently nor equally represented on each collecting day. Thus, a spider would appear to be more specialized, in comparison with its entire prey spectrum across the entire study, when observed at only one of these 93 time points. Hence, in our study the rate of prey detection used to fit a Poisson is a highly heterogeneous parameter, which complicates interpreting the deviation from a fitted Poisson. 

• The ability of multiplex PCR to detect several different prey items in a single predator sample depends upon many factors (3). We necessarily employed multiplex PCR to analyze the prey of 3300 spiders because 21 possible predator-prey interactions needed to be tested for each spider. Testing the effectiveness and sensitivity of our multiplex procedures for all combinations of prey and predators would have been unfeasible. Although our multiplex protocol did detect more than one prey item in several samples, it may not have been optimal for the wide range of potential prey for which we tested. 

In conclusion: Given the high diversity of spiders and potential prey, the extended collecting period, and the challenges of multiplex PCR, the low number of spiders for which two prey was detected is not as surprising as might first appear and is no cause for concern. Reliance on multiplex PCR was the only feasible way to examine each of 3300 spiders for 21 possible trophic interactions. Even if multiplex PCR had failed to uncover some joint prey occurrences, there is no reason to question its overall effectiveness in uncovering broad patterns of trophic interactions, as our prey detection rate is comparable to that of other studies. Following are two examples of the application of primer-based singleplex PCR to similar systems. In a study of predation by 1,231 spiders in one genus of Lycosidae in a North American forest, Whitney et al. (4) found that 44% tested positive for Collembola and 33% tested positive for Diptera. In a study of 128 spiders (representing 11 families) from two Russian forests, Zuev et al. (5) found that 17% of 968 possible interactions (128 x 6 different non-spider prey) tested positive for prey DNA. 

References

1. Wise DH. Spiders in Ecological Webs. Cambridge: Cambridge University Press; 1993. 328 p.

2. Foelix RF. Biology of Spiders. Second ed. Oxford: Oxford Univeristy Press; 2011. 432 p.

3. King RA, Read DS, Traugott M, Symondson WOC. Molecular analysis of predation: a review of best practice for DNA-based approaches. Molecular Ecology. 2008;17(4):947-63.

4. Whitney TD, Sitvarin MI, Roualdes EA, Bonner SJ, Harwood JD. Selectivity underlies the dissociation between seasonal prey availability and prey consumption in a generalist predator. Molecular Ecology. 2018;27(7):1739-48.

5. Zuev A, Heidemann K, Leonov V, Schaefer I, Scheu S, Tanasevitch A, et al. Different groups of ground-dwelling spiders share similar trophic niches in temperate forests. Ecological Entomology. 2020;45(6):1346-56.” 

End of S1 File -----------------------

10. 276 “we could not test for intra-family predation”; 

Are there any available data to give an idea of how important this is and how the absence of it influences the conclusions? I would assume there is a lot of intra-family predation.

We could not measure rates of intra-family predation because we did not utilize genus-specific primers. We now discuss intra-family predation in paragraphs added to the Discussion (given after Comment #13). 

11. 499-502 And can we learn something from this comparison?

The implications of the overall constancy in spider-spider predation rates are discussed later in the Discussion when we argue for the virtues of this food web as a model empirical system. We discuss the likelihood that spider-spider predation is greater than 50% in the paragraphs added to the Discussion (after Comment #13). ______________________________

12. 506-513 What do we know about seasonal variation in populations of these studied species, including non-spider ones? Their relative abundances certainly influence all the studied interactions and may help explain shifts in IGPrey.

We agree that such information would be very helpful, but collecting data on densities of spiders and their prey was beyond the intended scope of this project and would not have been feasible because we did not have the resources to collect quantitative data on seasonal changes, even over one year, in densities of spiders and potential prey. 

We prefer not to expand the generalizations contained in these lines. Detailed speculations on seasonal variation in densities of the genera in the 11 spider families and the 11 non-spider taxa, along with thorough citations of relevant literature, would more than double the length of the manuscript and, in addition to going well beyond the data, would detract from the major points that we wish to emphasize. 

13. 520-542 This section compares the results with Michalko´s paper, which I agree is important. However, I was expecting hypotheses to explain why the consistent grouping of IGP interactions by foraging mode (web vs cursorial). Is it possible to suggest reasons for this pattern?

This consistent grouping of IGP interactions, along with speculation on why rates of spider-spider predation are likely higher than we detected, are discussed in three new paragraphs in the Discussion, which now appear after the treatment of Michalko et al. (2022):

“The segregation of spider-spider predation into two channels in forests is likely more pronounced than our results suggest. Non-sexual cannibalism occurs widely among all instars in numerous spider species (48). Although cannibalism is more frequent between stages that differ substantially in size, cannibalism among recently hatched spiderlings can be a major mortality factor for lycosids on the forest floor (49). Intra-family predation may even be more prevalent than cannibalism, because spiders within the same genus, and often in different genera within the same family, exhibit similar prey-foraging behaviors. We could not detect within-family predation because we did not use genus-level primers, and using standard DNA techniques to detect cannibalism in natural populations is not feasible because prey and predator DNA sequences are too similar. The likely widespread occurrence of cannibalism and intra-family predation makes it is reasonable to hypothesize that much more than 90% of spider-spider predation on the forest floor is restricted to one of two interaction channels defined by foraging mode. 

Although this pattern supports the hypothesis that susceptibility to attack by other spiders relates directly to behaviors that are critical to foraging for prey, other factors must play a role. Web spinners should be exposed to predation when searching for new web sites, and when searching for mates – why, then, do they not appear more often in the diets of cursorial spiders? Species of several web-spinning families of the ground layer also forage for prey off the web (16) -- should not that behavior also expose them to predation by cursorial families? Cursorial spiders may be able to avoid webs, but why do they not fall prey more frequently to those web spinners that also forage off the web? Coupled with these questions is the finding that predation between web spinners and cursorial spiders may be more frequent in habitats other than the forest floor (47). Furthermore, if the gut DNA of juvenile spiders on the forest floor were to be analyzed, inter-channel predation might turn out to occur more -- or maybe less -- often than our focus on adult predators revealed. Clarifying the boundaries of these two interaction channels awaits more analyses of spider prey at the species level for a wider spectrum of spider life stages. Recently implemented NGS barcoding techniques with spiders (44, 45, 50-52) promise major insights if applied to these two interaction channels in spider-focused food webs.”

Reviewer #2:

14. How many samples were collected per day at each location?

The number varied according to the number of searchers. We collected ~14,000 spiders over 93 days (these numbers are given in the manuscript), yielding a rough average of about 150 spiders collected per day. We point out in the manuscript how we selected the 3300 spiders to analyze for prey DNA.

15. The study was conducted in a 320 ha preserve. How was the site selected for leaf litter collection and sorting? For example, was the leaf litter collected in 50cm x 50cm plots?

No, we did not sample quantitatively. Instead, we searched systematically over a large undefined area each day.

16. Is it possible to estimate the total area that was sampled based on the total size of the litter collection plots?

No. Our goal was not to obtain density data.

17. The sampling was done between 10:00 and 16:00. Can I assume that the spiders forage more at night? If so, could we expect variation 

in spider abundance or variation in guild or family composition?

Not all spiders that we collected limit their foraging to nighttime, nor necessarily forage more at night. We actively searched and sifted the litter. Spiders that did forage at night in the litter would still be collected, although some may have stayed in their retreats and were missed.

18. It would be interesting to see images of the study area and perhaps a figure or map showing the shape of the reserve and the distribution of leaf litter collection sites in the area.

We could provide images of the study area, but adding them would lengthen the manuscript without adding any helpful details because the forest did not differ in aspect or structure from similar forests. We prefer to keep the manuscript as concise as possible.

Providing a detailed map of precise locations of collecting areas is not feasible, and any approximation might require several maps, and in the end, would not help interpret our findings. This fact may be more apparent now that we have explained our collecting protocol more fully. 

General response to questions and comments of Reviewer #2:

The above questions reflect a misunderstanding (also shared with Reviewer #1) of our systematic collecting procedures due at least partly to our frequent use of the term “sampling” instead of “collecting.” The two extensively revised initial paragraphs of the “Materials and methods” section (see response to Comment #5 above) should clear up confusion about whether our collecting methods could be used to estimate spider densities.

---

## [Decision Letter · Decision Letter 1]

9 Oct 2023

Pattern of seasonal variation in rates of predation between spider families is temporally stable in a food web with widespread intraguild predation

PONE-D-23-17199R1

Dear Dr. Matthew A. McCary

We’re pleased to inform you that your manuscript has been judged scientifically suitable for publication and will be formally accepted for publication once it meets all outstanding technical requirements.

Kind regards,

Kleber Del-Claro, PhD

Academic Editor

PLOS ONE

Additional Editor Comments (optional):

Reviewers' comments:

Reviewer's Responses to Questions

**Comments to the Author**

1. If the authors have adequately addressed your comments raised in a previous round of review and you feel that this manuscript is now acceptable for publication, you may indicate that here to bypass the “Comments to the Author” section, enter your conflict of interest statement in the “Confidential to Editor” section, and submit your "Accept" recommendation.

Reviewer #1: All comments have been addressed

Reviewer #2: All comments have been addressed

2. Is the manuscript technically sound, and do the data support the conclusions?

Reviewer #1: Yes

Reviewer #2: Yes

3. Has the statistical analysis been performed appropriately and rigorously? 

Reviewer #1: Yes

Reviewer #2: Yes

4. Have the authors made all data underlying the findings in their manuscript fully available?

Reviewer #1: Yes

Reviewer #2: Yes

5. Is the manuscript presented in an intelligible fashion and written in standard English?

Reviewer #1: Yes

Reviewer #2: Yes

6. Review Comments to the Author

Reviewer #1: This is the revised version. I carefully read all the answers and the modifid version. My comments have all been adequately addressed

Reviewer #2: Dear Editor,

I am sending you my comments on the manuscript entitled "Pattern of seasonal variation in rates of predation among spider families is temporally stable in a food web with widespread intraguild predation" (PONE -D-23-17199R1).

First, I would like to thank you for the opportunity to review this manuscript. I am pleased to report that questions and concerns I raised in my initial review have been adequately addressed. The clarity of the text, particularly with respect to the spider collection procedures, has improved significantly, making it more accessible to a broader readership.

In addition, I find the study not only innovative, but also very appealing to a broad audience. The observed temporal stability of predation rates among spider families adds a valuable dimension to our understanding of intraguild predation dynamics.

Therefore, I have no further comments or suggestions on the new manuscript version.

Thank you again for the opportunity to participate in this peer review process.

7. PLOS authors have the option to publish the peer review history of their article (what does this mean?). If published, this will include your full peer review and any attached files.

Reviewer #1: No

Reviewer #2: No

---

## [Editor Report · Acceptance letter]

19 Oct 2023

PONE-D-23-17199R1 

Pattern of seasonal variation in rates of predation between spider families is temporally stable in a food web with widespread intraguild predation 

Dear Dr. McCary:

I'm pleased to inform you that your manuscript has been deemed suitable for publication in PLOS ONE. Congratulations! Your manuscript is now with our production department. 

Kind regards, 

on behalf of

Dr. Kleber Del-Claro 

Academic Editor

PLOS ONE